# Triplon current generation in solids

Yao Chen [1], Masahiro Sato[2 ✉], Yifei Tang[1], Yuki Shiomi[3], Koichi Oyanagi [1,4], Takatsugu Masuda [5], Yusuke Nambu [1,6,7], Masaki Fujita [1] & Eiji Saitoh[1,8,9,10,11 ✉]

A triplon refers to a fictitious particle that carries angular momentum $S=1$ corresponding to the elementary excitation in a broad class of quantum dimerized spin systems. Such systems without magnetic order have long been studied as a testing ground for quantum properties of spins. Although triplons have been found to play a central role in thermal and magnetic properties in dimerized magnets with singlet correlation, a spin angular momentum flow carried by triplons, a triplon current, has not been detected yet. Here we report spin Seebeck effects induced by a triplon current: triplon spin Seebeck effect, using a spin-Peierls system $CuGeO_3$. The result shows that the heating-driven triplon transport induces spin current whose sign is positive, opposite to the spin-wave cases in magnets. The triplon spin Seebeck effect persists far below the spin-Peierls transition temperature, being consistent with a theoretical calculation for triplon spin Seebeck effects.

[1] Institute for Materials Research, Tohoku University, Sendai, Japan. [2] Department of Physics, Ibaraki University, Mito, Ibaraki, Japan. [3] Department of Basic Science, The University of Tokyo, Tokyo, Japan. [4] Faculty of Science and Engineering, Iwate University, Morioka, Japan. [5] Institute of Solid State Physics, The University of Tokyo, Kashiwa, Chiba, Japan. [6] FOREST, Japan Science and Technology Agency, Kawaguchi, Saitama, Japan. [7] Organization for Advanced Studies, Tohoku University, Sendai, Japan. [8] Department of Applied Physics, The University of Tokyo, Tokyo, Japan. [9] Institute for AI and Beyond, The University of Tokyo, Tokyo, Japan. [10] Advanced Institute for Materials Research, Tohoku University, Sendai, Japan. [11] Advanced Science Research Center, Japan Atomic Energy Agency, Tokai, Japan. ✉email: masahiro.sato.phys@vc.ibaraki.ac.jp; eizi@ap.t.u-tokyo.ac.jp

Spin Seebeck effects[1] (SSEs) refer to the generation of a spin current, a flow of spin angular momentum of electrons, from a temperature gradient applied to a bilayer system comprising a magnet and heavy metal such as Pt. Spin current in the magnet propagates along the temperature gradient and reaches the heavy metal[2], in which spin current can be detected as a voltage signal via the inverse spin-Hall effect (ISHE)[3,4]. The SSE has been observed in various insulating systems, in which parasitic thermal effects originating from itinerant electrons can be eliminated. Some types of spin carriers including magnons in magnets[5–8], paramagnons[9], antiferromagnetic magnons[7,8], and spinons in a spin liquid[10,11] have been demonstrated for different mechanisms of the SSE.

Among quantum spin systems without magnetic order, dimerized magnets spin systems occupy an important position, in which two neighboring spins are frozen as $S = 0$ singlets in the ground state. The elementary spin excitations are $S = 1$ triplet states, called a triplon. A triplon has been expected to carry spin angular momentum.

In this work, we report the observation of spin current in CuGeO$_3$, carried by triplons in terms of longitudinal SSE measurements. A typical spin-dimer system is a spin-Peierls material CuGeO$_3$, which contains one-dimensional spin-1/2 chains with antiferromagnetic exchange interaction for nearest-neighbor spins. In the spin chain, with lowering the temperature, neighboring spins dimerizes to form a spin-gapped phase; the transition is called a spin Peierls (SP) transition[12]. Below the SP transition temperature $T_{SP}$, the chain distorts so that the distances between neighboring spins change alternately. The bond-alternating exchange interaction causes neighboring spins to dimerize to reduce the total energy, creating a gap in the spin excitation energy spectrum. CuGeO$_3$ is the firstly discovered inorganic system that exhibits a SP transition[13]. High-quality single crystals of CuGeO$_3$ are easier to obtain than organic SP materials and CuGeO$_3$ has typically been used to study spin excitations in SP systems[14].

## Results

**Sample structure and the concept of the study.** Figure 1a shows the crystal structure of CuGeO$_3$. One-dimensional Cu$^{2+}$ spin ($S = 1/2$) chains are aligned along the $c$-axis, as illustrated as dotted lines. The nearest-neighbor exchange interaction along the $c$-axis is $J_c \sim 120$ K, much greater than the interchain exchange coupling[15], $J_b \sim 0.1 J_c$, and $J_a \sim -0.01 J_c$ along the $b$ and $a$-axis, respectively. CuGeO$_3$ is thus considered as a quasi-one-dimensional spin system. As the temperature $T$ decreases below the SP transition temperature $T_{SP} \sim 14.5$ K, the lattice of CuGeO$_3$ is spontaneously distorted to form a bond alternating configuration[16]. The spin configuration in the ground state is shown schematically in Fig. 1b, where all spins are frozen as dimerized $S = 0$ singlets. The elementary spin excitation from the ground state is an $S = 1$ triplon[17], also shown in Fig. 1b. The excitation gap of a triplon is estimated to be ~23 K from ESR[18] and neutron scattering[15,19,20] experiments.

Based on the previous neutron scattering results[19] and theoretical calculations[21], the dispersion relation of triplons along the $c$-axis is sketched in Fig. 1c. In the absence of a magnetic field, $\mu_0 H = 0$, the triplon states are threefold degenerated: $|\uparrow\uparrow\rangle$, $(|\uparrow\downarrow\rangle + |\downarrow\uparrow\rangle)/\sqrt{2}$, and $|\downarrow\downarrow\rangle$ corresponding to states with different spin quantum numbers, $S_z = 1, 0$, and $-1$, respectively. When a magnetic field is applied, the degenerated triplet bands split into three due to the Zeeman effect, where the triplet state with the parallel spin direction to $H$, $|\uparrow\uparrow\rangle$, has the lowest energy[18–20]. The $|\uparrow\uparrow\rangle$ state thus exhibits the highest probability

for thermal excitation, while $|\downarrow\downarrow\rangle$ the lowest. Driven by a thermal gradient, different occupancy among the three states can result in a flow of a net spin angular momentum in an external magnetic field: triplon SSE. One of the key features of the triplon SSE is that it should exhibit the opposite sign of ISHE signal to the magnon SSE in ferri- and ferro- magnets. This is because triplon spin currents are carried by excitation with the parallel spin direction to the external field; while magnons carry antiparallel spins to the external field, as shown in Fig. 1d. In these two systems, the interfacial accumulated spins in the magnets are injected into the metal (Pt) through the interfacial exchange interaction. Since the exchange interaction conserves the spin angular momentum, the spin current in the metal has the same polarization direction as that in the magnets. As a result, the sign of ISHE in the triplon SSE is opposite to that in the magnon SSE. Our theoretical calculation based on the tunnel spin current (see Supplementary Note G3 for details) suggests that the opposite sign of triplon SSE to magnon SSE is a universal property for gapped triplet-spin systems.

**Material characterization and measurement setup.** First, we measured the temperature dependence of magnetization, $M(T)$, for the CuGeO$_3$ sample. Figure 1e shows the magnetic susceptibility $\chi(T) = M(T)/H$ measured at $\mu_0 H = 1$ T $\|b$-axis. When $T$ is decreased from 20 K, $\chi$ exhibits a sudden drop at the SP transition temperature $T_{SP} \sim 14.5$ K and rapidly decreases towards zero. This indicates that Cu spins along the spin chain ($\|c$-axis) form dimers and a spin gap develops below $T_{SP}$. By measuring $\chi(T)$ at different values of $H$ (see Supplementary Fig. 1 for details), we obtain the $H$–$T$ spin-phase diagram of the CuGeO$_3$ sample in Fig. 1f. In the phase diagram, for large $H$ in the SP phase, CuGeO$_3$ undergoes another phase transition into a magnetic phase at a transition field, $H_m$, where magnetization is recovered by forming a spin soliton lattice in the chain[22]. The transition can be confirmed as a sudden jump of magnetization as a function of fields[23] (also see Supplementary Fig. 1). The obtained phase diagram is in good agreement with a previous study[24].

The experimental setup for the SSE measurement and the sample used in the present study are shown in Fig. 2a, b. A trilayer structure comprising a Pt wire for ISHE, an insulating SiO$_2$ layer, and an Au wire as a heater was fabricated on the top of the (001) plane of a CuGeO$_3$ single crystal. The spin Seebeck voltage signal $V_{SSE}$ in the Pt wire is measured by a lock-in method, where an a.c. current is applied to the Au layer under an in-plane magnetic field, $H$, applied perpendicular to the Pt wire. We normalize $V_{SSE}$ by the heater power and the detector resistance as $\tilde{V}_{SSE} = V_{SSE}/(P_{Au} \cdot R_{Pt})$.

**Observation of the triplon SSE.** The magnetic field dependence of $\tilde{V}_{SSE}$ at some selected temperatures is shown in Fig. 2c, d. At 15 K, just above $T_{SP} \sim 14.5$ K, no $H$-dependent signal is recognized. Most notably, a clear voltage signal appears when $T$ is decreased down to 2 K. The signal is an odd function of $H$, reflecting the symmetry of the ISHE[25]. The sign of the $\tilde{V}_{SSE}$ signal is opposite to that of the magnon-mediated SSE[26,27], which is confirmed in a similar setup for a ferrimagnetic insulator Y$_3$Fe$_5$O$_{12}$/Pt sample in Fig. 2e, f. The result indicates that spin current carriers in these two systems have different spin polarization directions with respect to $H$, consistent with the triplon-current senario. Note that the sign of the SSE signals in spin nematic LiCuVO$_4$/Pt (ref. [28]) is also opposite to the present CuGeO$_3$/Pt case. We also note that, unlike the magnon-mediated SSE in ferro/ferrimagnets, the triplon-mediated SSE does not scale with the $M(H)$ curve. For the magnon-mediated SSE[27], $M(H)$ represents the increase in the spin-current polarization. In

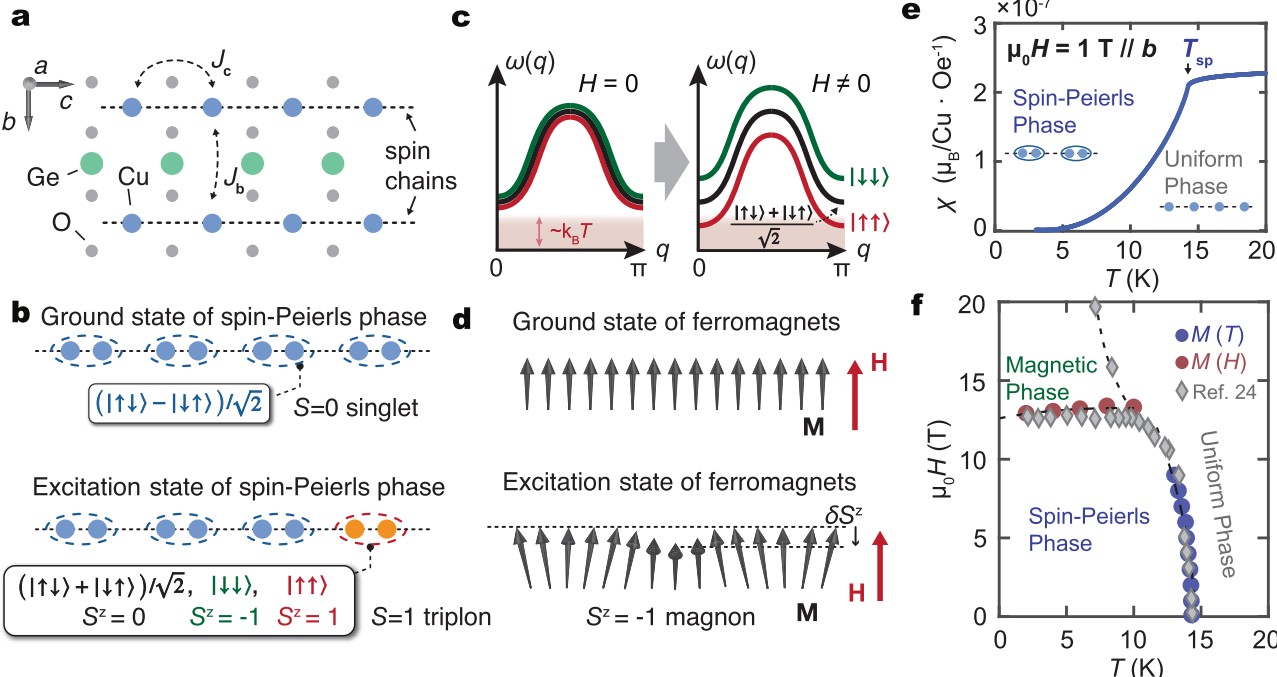

**Fig. 1 Spin chain and spin-Peierls (SP) phase of CuGeO₃. a** Crystalline structure of $CuGeO_3$. The Cu spin chains are shown as dotted lines. Spin-1/2 spin chains along the $c$-axis in $CuGeO_3$ are formed by staking $CuO_2$ chains. $J_c$ and $J_b$ denote the exchange interactions along the $c$ and $b$-axis, respectively. **b** When temperature ($T$) falls down to the SP transition temperature $T_{SP}$, $CuGeO_3$ undergoes a SP transition. After the transition, the Cu spins dimerize and the ground state becomes a spin singlet state $((|\uparrow\downarrow\rangle - |\downarrow\uparrow\rangle)/\sqrt{2}$ with $S = 0$). Elementary excitation in the SP phase is a triplon with $S = 1$. **c** Schematic band structure of triplon in $CuGeO_3$[19,21]. Three triplet states ($|\downarrow\downarrow\rangle$, $(|\uparrow\downarrow\rangle + |\downarrow\uparrow\rangle)/\sqrt{2}$ and $|\uparrow\uparrow\rangle$) are threefold degenerated at zero fields. By applying a magnetic field ($H$), the Zeeman effect lift the degeneracy. **d** The ground state and excitation state of a ferromagnet. Elementary excitation is a magnon with $S = 1$. The magnons reduce the magnetization (**M**) along the magnetic field (**H**) as $\delta S^z$. **e** $T$ dependence of magnetic susceptibility ($\chi$) of $CuGeO_3$ under an applied magnetic field of 1 T along the $b$-axis. The SP transition temperature $T_{SP}$ is illustrated as the arrow in the figure. **f** Magnetic field - temperature ($H$-$T$) spin phase diagram for the $CuGeO_3$ sample obtained from the $M(H)$ and $\chi(T)$ measurement results. The diamond symbols represent data taken from a previous study[24]. The dotted curves are guide for the eyes.

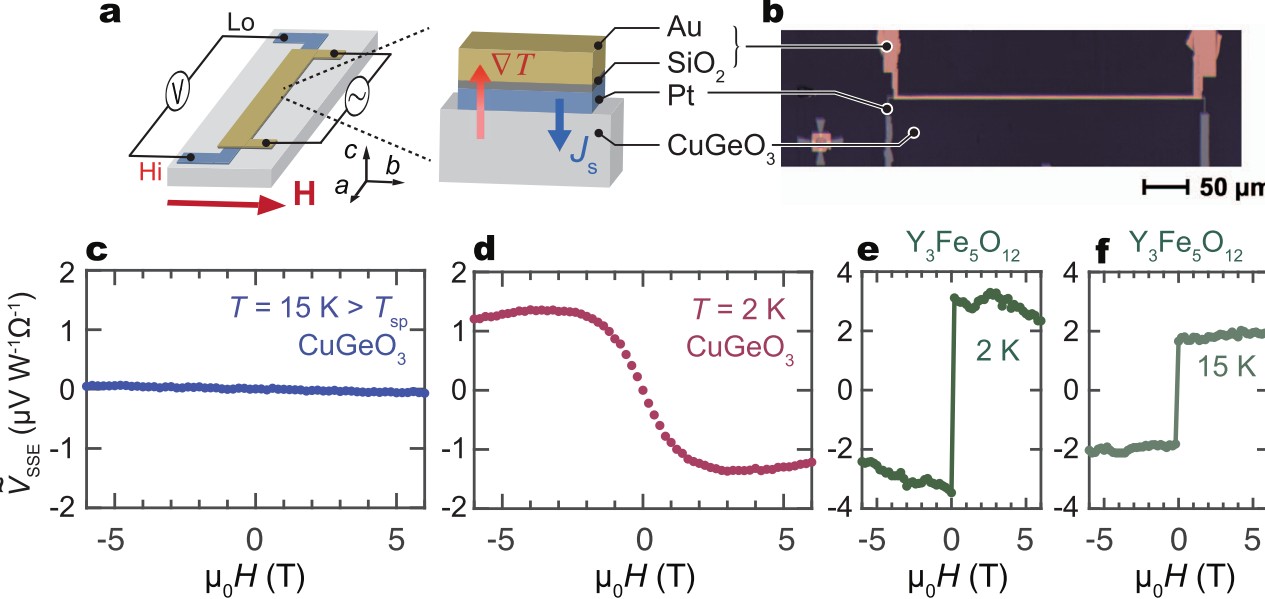

**Fig. 2 SSE measurement results for CuGeO₃/Pt. a** Experimental setup for the SSE measurement. Temperature gradient ($\nabla T$) is applied by using an Au heater electrically insulated from the Pt layer by a $SiO_2$ film. An a.c. current is applied to the Au heater and SSE voltage in the Pt is measured with a lock-in amplifier as the second harmonic voltage $V_{2f}$. **b** An optical micrograph for a on-chip SSE device made on the top of $CuGeO_3$. **c**, **d** The magnetic field ($H$) dependence of $\tilde{V}_{SSE}$ (the SSE signal normalized to the heating power and the detector resistance) for the $CuGeO_3$/Pt sample at 15 and 2 K, respectively. **e**, **f** $\tilde{V}_{SSE}(H)$ measured for $Y_3Fe_5O_{12}$/Pt at 2 and 15 K.

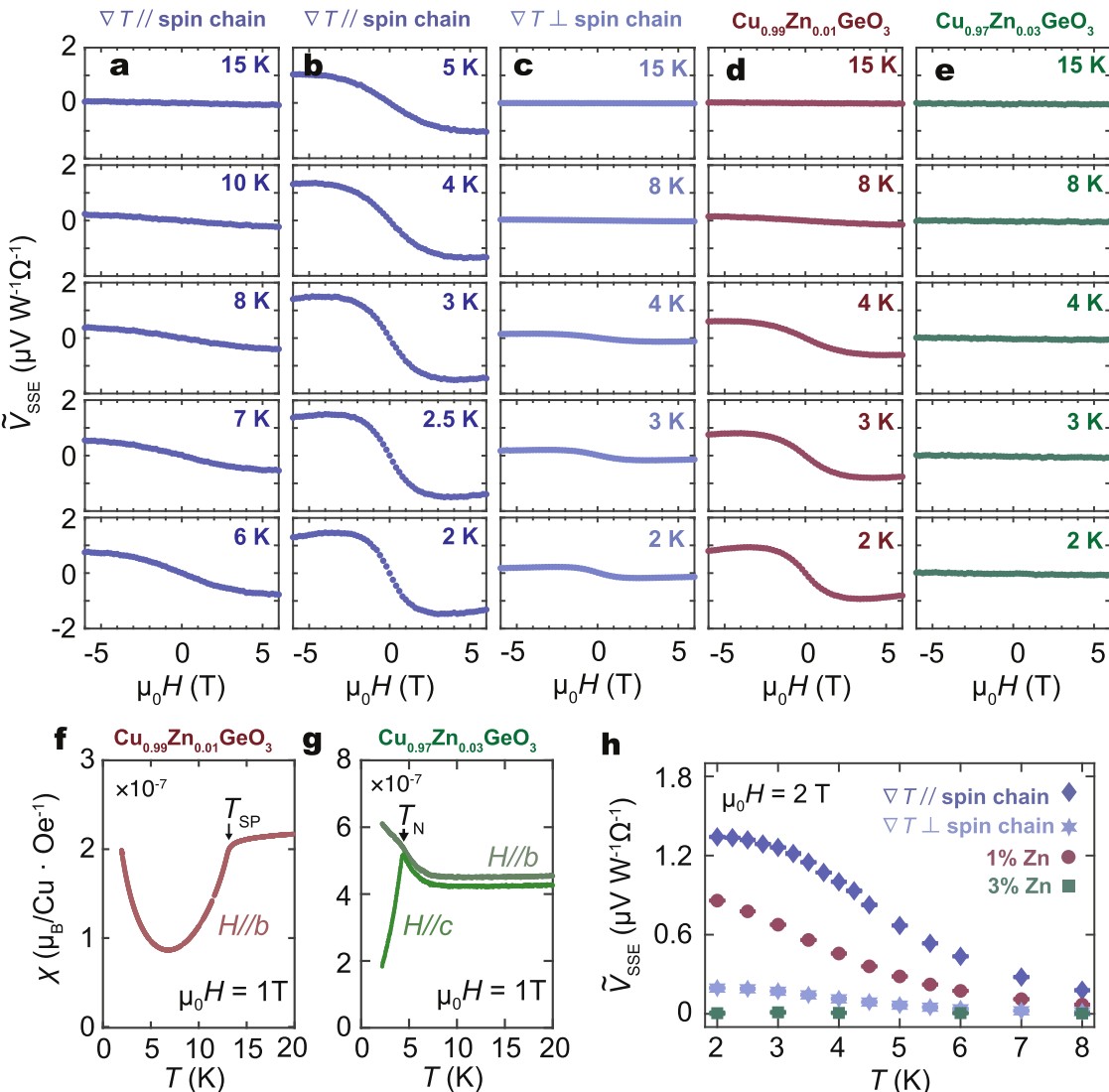

**Fig. 3 Temperature dependence of SSE signal. a, b** Magnetic field ($H$) dependence of $\tilde{V}_{SSE}$ (the SSE signal normalized to the heating power and the detector resistance) at selected temperatures ($T$) for CuGeO$_3$/Pt, with temperature gradient ($\nabla T$) applied along the spin chain. **c** $H$ dependence of $\tilde{V}_{SSE}$ for CuGeO$_3$/Pt, with $\nabla T$ applied perpendicular to the spin chain. **d, e** $H$ dependence of $\tilde{V}_{SSE}$ at selected temperatures for Cu$_{0.99}$Zn$_{0.01}$GeO$_3$/Pt and Cu$_{0.97}$Zn$_{0.03}$GeO$_3$/Pt, respectively. In **a–e**, the SSE signal is normalized by the heating power and the detector resistance. **f, g** $T$ dependence of magnetic susceptibility for Cu$_{0.99}$Zn$_{0.01}$GeO$_3$ and Cu$_{0.97}$Zn$_{0.03}$GeO$_3$, respectively. **h** $T$ dependence of SSE signal in different samples at $\mu_0H = 2$ T. Error bars represent standard deviation.

contrast, the magnetization increase in the SP phase means larger density of broken dimers and the scattering probability between triplons and broken dimers becomes larger. The unusual $H$-dependence of the triplon SSE originates from the different properties between ferro/ferrimagnets and SP phases, and it is one of the characteristics of triplon SSE. (see Supplementary Fig. 2 for details).

As shown in Fig. 3a, b, the temperature dependence of $\tilde{V}_{SSE}$ shows that the signal magnitude increases monotonically as $T$ decreases from $T_{SP}$ to 3 K. When $T < 3$ K, the SSE voltage remains almost unchanged down to 2 K. At higher temperatures, the dimerized ground state is weakened because the thermally excited phonon disturbs the spontaneous displacement of the Cu$^{2+}$ ion. In other words, the singlet ground state and triplon excitation is no longer a good picture for the low-energy excitation, and the triplon lifetime shrinks toward higher temperatures and completely disappears at $T_{SP}$. This effect has been observed as an increase in the peak width of the triplon excitation in neutron scattering experiments[29]. We note that, by phenomenologically

incorporating the triplon-phonon scattering term in a calculation, the $T$ dependence of $\tilde{V}_{SSE}$ at moderate temperatures can be explained qualitatively (see Supplementary Note G4 for details).

**Influence of crystal orientation and Zn-doping on the triplon SSE.** Let us turn to the $H$ dependence of $\tilde{V}_{SSE}$ in CuGeO$_3$/Pt. In the entire temperature range, as $H$ increases from zero, $\tilde{V}_{SSE}$ increases first and takes its maximum at $\mu_0H \sim 2$ T. As $H$ further increases, $\tilde{V}_{SSE}$ is then suppressed. As shown theoretically in the following, triplon scattering by inevitable impurities and defects in the nominally pure CuGeO$_3$ sample can explain the observed $H$ dependence of $\tilde{V}_{SSE}$.

To show that the observed voltage is related to the triplon in the SP phase, we performed another control experiment with $\nabla T$ applied perpendicular to the spin chain configuration. Figure 3c shows $\tilde{V}_{SSE}(H)$ for $\nabla T \perp c$ in a reference sample prepared with the same procedure as $\nabla T \| c$. The observed SSE signal in $\nabla T \perp c$ is only 1/7 in magnitude as compared to the $\nabla T \| c$ configuration,

consistent with the highly anisotropic transport of triplons. This result corroborates that the origin of the observed signal is the 1-D triplon transport. Moreover, additional reference experiments are performed for two nonmagnetic Zn-doped $CuGeO_3$ samples, in which the triplon spin current should be suppressed by Zn-induced scattering. Figure 3d, e show the $\tilde{V}_{SSE}(H)$ for $Cu_{0.99}Zn_{0.01}GeO_3/Pt$ and $Cu_{0.97}Zn_{0.03}GeO_3/Pt$ in the same temperature range as the $CuGeO_3/Pt$ measurements. Clearly, $\tilde{V}_{SSE}$ is suppressed in the 1% Zn-doped sample and disappeared in the 3% Zn-doped sample.

Figure 3f, g shows $\chi$ for $Cu_{0.99}Zn_{0.01}GeO_3$ and $Cu_{0.97}Zn_{0.03}$-$GeO_3$, respectively. For $Cu_{0.99}Zn_{0.01}GeO_3$, the SP transition is observed at 13.2 K. The decrease of $T_{SP}$ in doped $CuGeO_3$ agrees with the result from the previous reports[30]. On the other hand, the SP transition is not visible in $\chi(T)$ in $Cu_{0.97}Zn_{0.03}GeO_3$. Instead, a clear cusp is observed in $\chi_c(T)$ (with $H\|c$-axis) at $T_N \sim 4.3$ K, indicating phase transition to an antiferromagnetic phase. The $T_N$ is consistent with the previously reported value for Zn-3% doped $CuGeO_3$ (ref. [30]). The doping effect implies that non-magnetic Zn breaks the Cu-spin chains and introduces unpaired Cu free spins, called solitons, into the spin chain. Solitons disturb the dimerization, and local antiferromagnetic correlation may lead to long-range antiferromagnetic order[31] at low temperatures in $Cu_{0.97}Zn_{0.03}GeO_3$. As shown in Fig. 3h, $\tilde{V}_{SSE}$ is significantly suppressed in the $\nabla T\perp$ spin-chain configuration and the Zn-doped $CuGeO_3$ samples in all $T$ range. In the doped samples, the spin-excitation gap is still present in the SP phase, but their transport may be blocked by Zn. The result suggests that the triplon spin current is strongly influenced by Zn-induced scattering. Curie-Weiss fitting of $\chi(T)$ at low $T$ (<3 K)[13,32] (see Supplementary Note F for details) reveals that the density of free $S = 1/2$ spins is estimated to be 0.02% in the nominally pure $CuGeO_3$ sample, so the effect of triplon scattering by impurities cannot be ignored. Therefore, we attempted to formulate the triplon spin current in the manner of Boltzmann equation, which is capable of incorporating scattering effects.

**Theoretical model for the triplon SSE.** Figure 4 compares the $H$ dependence of the observed SSE voltage signal with theoretical results. As we already discussed, $|\tilde{V}_{SSE}|$ is expected to monotonically grow with increasing $H$ because the triplon density of $S_z = 1$, i.e., the dominant carrier of spin current, increases with lowering the spin gap up to the critical field $H_m$. However, the observed broad peak (see Fig. 4a) and the Zn-doped results shown in Fig. 3 strongly suggest the importance of triplon

scattering processes. To theoretically analyze triplon currents in $CuGeO_3$, we here apply the Boltzmann equation[33]. The spin current $J_s$ computed by the Boltzmann equation (See Supplementary Note G2) is given by

$$J_s = \sum_{S_z} \int \frac{k}{2\pi} \hbar S_z (v_{S_z}(k))^2 \tau_k k_B \Delta T \frac{\epsilon_{S_z}(k)}{(k_B T)^2} \frac{\exp(\epsilon_{S_z}(k)/k_B T)}{\left(\exp\left[\epsilon_{S_z}(k)/k_B T\right] - 1\right)^2},$$

(1)

where $\Delta T$ is the temperature difference along the spin chain direction. $v_{S_z}(k)$, $\tau_k$ and $\epsilon_{S_z}(k)$ are the group velocity, relaxation time, and energy dispersion for the triplons[17], respectively. We assume that the triplon can be viewed as a bosonic particle and the final part of Eq. (1) stems from the Bose distribution function. The $S_z = 0$ branch of the triplon does not contribute to the spin current. At $H = 0$, the distribution functions of $S_z = \pm 1$ branches are identical, which cancel out the total spin current. By varying $T$ and $H$, the thermal excitation and the Zeeman effect alter the triplon distributions. Assuming a constant $\tau_k$, the population difference between $S_z = \pm 1$ increases monotonically at a fixed $T(<T_{SP})$ due to the Zeeman effect, resulting in an almost $H$-linear increase in $J_s$. The $H$-linear behavior can also be reproduced by using a tunnel spin-current formula[10,28,34,35], which is another microscopic theory for thermal spin current (see Supplementary Note G3).

As shown in Fig. 4a, $|\tilde{V}_{SSE}(H)|$ takes a maximum at around 3 T. To explain the non-monotonic behavior of $\tilde{V}_{SSE}$, we considered two mechanisms of triplon scattering processes. One is the scattering by nonmagnetic impurities, where the scattering probability $1/\tau_{k,\text{non-mag}}$ is constant with respect to $H$. Another source of scattering is free spins of broken dimers and unpaired $Cu^{2+}$ ($S = 1/2$). The density of scatterers and scattering potential are assumed to depend on $H$ as a Brillouin function. The total scattering probability is then the sum of the two scattering mechanisms, $1/\tau_{k,\text{total}} = 1/\tau_{k,\text{non-mag}} + 1/\tau_{k,\text{mag}}$. Assuming that the scattering effect by magnetic impurities is dominant, we can reproduce the experimentally obtained $H$ dependence of $\tilde{V}_{SSE}$ shown in Fig. 4b. In the low $H$ regime, where the spin of magnetic impurities is not aligned and the scattering effect is weak, the Zeeman effect dominates the generation and transport of the triplon current. With increasing $H$, the triplon mode of $S_z = +1$ ($S_z = -1$) shifts downward (upward) due to the Zeeman effect, and the imbalance between $S_z = \pm 1$ triplons becomes larger, resulting in an increase in the total triplon current. As $H$ is further increased, the impurities are magnetized and magnetic scattering

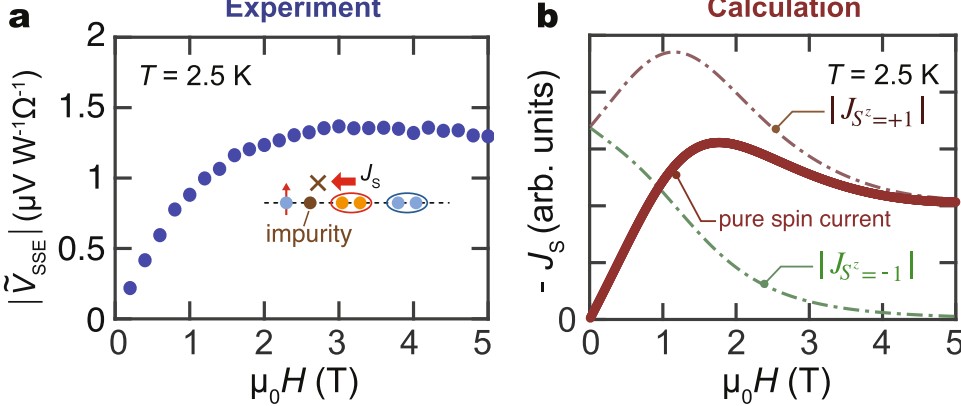

**Fig. 4 Magnetic field dependence of SSE voltage. a** Magnetic field ($H$) dependence of the magnitude of $|\tilde{V}_{SSE}|$ (absolute value of the SSE signal normalized to the heating power and the detector resistance) at 2.5 K. **b** Calculation results of the $H$ dependence of the triplon spin current (see text and Supplementary Note G2 for details).

dominates the triplon transport, resulting in a broad peak as shown in Fig. 4b. The result agrees with the experimentally obtained data shown in Fig. 4a at a semi-quantitative level.

## Discussion

Finally, we here examine other possible origins for the observed thermoelectric voltage. One candidate could be the paramagnetic SSE due to free Cu spins. Several papers have addressed the spin current transport in paramagnets, including $Gd_3Ga_5O_{12}$ (refs. [9,36]), $DyScO_3$ (ref. [9]), and $La_2NiMnO_6$ (ref. [37]). Paramagnon is believed to be responsible for spin transport in paramagnets as a result of short-range magnetic correlation or long-range dipole interactions. The spin diffusion length of $Gd_3Ga_5O_{12}$ is estimated to be about 1.8 µm at 5 K[36]. However, in our $CuGeO_3$ sample with the free spin density of 0.02%, the average distance between free spins is estimated to be around 0.29 nm/0.02% ~ 1.5 µm (the distance between neighboring $Cu^{2+}$ ions along the $c$-axis is 0.29 nm[38]). This means that the paramagnetic spins are too dilute for spin correlation and spin current transport. Furthermore, the suppression of $V_{SSE}$ in the Zn-doped samples, which exhibit larger paramagnetic moments than $CuGeO_3$, also rules out the possibility of paramagnetic SSE.

The normal and anomalous Nernst effects may also contribute to an electric voltage under a temperature gradient, but these effects contradict with the suppression of the signal in the high-temperature spin-liquid phase; the vanishing signals in the $\nabla T \perp c$ configuration and the Zn-doped samples (see Fig. 3a–e). Furthermore, the normal Nernst effect should be linear with respect to the applied magnetic field and can not explain the observed magnetic field dependence of $V_{SSE}$ shown in Fig. 3a, b.

Note that the voltage observed at 15 K in the $CuGeO_3$/Pt shows almost no $H$-dependence, as shown in Fig. 2c. Above $T_{SP}$, the elementary excitation of spin chains are gapless spinons[21]. A spinon spin current, a spin current carried by spinons, was found in $Sr_2CuO_3$ (ref. [10]). The spinon spin current may contribute to a small $H$-linear voltage signal above $T_{SP}$ (see Supplementary Note C for details). In the spin-Peierls phase, the gapless spinon is replaced by the gapped triplon[21].

In summary, we observed the triplon spin-Seebeck effect in $CuGeO_3$/Pt. Due to the triplon excitation in the system, the sign of the observed SSE is opposite to that of the conventional magnon SSE. The triplon SSE signal persists down to 2 K in the spin-Peierls phase and the magnetic field dependence of the triplon SSE is consistent with microscopic calculation. Our result shows that the spin-Seebeck effect also acts as a probe for spin excitations in gapped spin systems, and can also be applied to other materials with exotic spin excitation, such as a spin ladder system $SrCu_2O_3$ (ref. [39]) and a spin dimer system $SrCu_2(BO_3)_2$ (ref. [40]).

## Methods

**Sample fabrication**. $CuGeO_3$ single crystals were grown by a traveling solvent floating zone (TSFZ) method. Samples were cut into size of ~7 mm × 3 mm for further measurements. 40nm-thick ferrimagnetic insulator $Y_3Fe_5O_{12}$ single crystals were grown on $Gd_3Ga_5O_{12}$(111) substrates by magnetron sputtering. Prior to the deposition, $Gd_3Ga_5O_{12}$ substrates were annealed in the air at 825 °C for 30 min in a face-to-face configuration. To crystalize the as-grown amorphous $Y_3Fe_5O_{12}$, the samples were post-annealed in the air at 825 °C for 200 s. For SSE measurements, on-chip devices with Pt detector and $SiO_2$/Au heater were fabricated on top of samples (both pure/doped $CuGeO_3$ and $Y_3Fe_5O_{12}$) by electron-beam lithography, magnetron sputtering and lift-off technique (see Supplementary Note A).

**SSE measurement**. We adapted the on-chip heating method for SSE, as illustrated in Fig. 1b. The resistance between the Au and Pt layers is much greater than MΩ at room temperature. A longitudinal temperature gradient was generated by applying an a.c. current with the frequency of $f = 13$ Hz to the Au heater. Since the SSE is driven by Joule heating, the frequency of the SSE signal on the Pt wire is $2f$. The SSE signal was detected via a lock-in method.

**Magnetization measurement**. The magnetization of $CuGeO_3$ was measured in a Quantum Design Physical Properties Measurement System (PPMS) with a vibrating sample magnetometer (VSM) option.

## Data availability
The data that support the findings of this study are available from the corresponding authors upon reasonable request. Source data are provided with this paper.

## Code availability
The codes used in theoretical simulations and calculations are available from the corresponding authors upon reasonable request.

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

## Acknowledgements

We thank D. Hirobe, T. Kikkawa, G. E. W. Bauer, and Y. Hirayama for valuable discussions. This research was supported by JST ERATO Spin Quantum Rectification Project (No. JPMJER1402), JPSP KAKENHI (the Grant-in-Aid for Scientific Research (S) No. JP19H05600 and JP17H06137; the Grant-in-Aid for Scientific Research (A) No. JP16H02125; the Grant-in-Aid for Scientific Research (B) No. JP20H01830 and No. JP19H02424; the Grant-in-Aid for Scientific Research (C) No. JP17K05513; the Grant-in-Aid for Challenging Research (Exploratory) No. JP19K22124 and No. JP20K20896; the Grant-in-Aid for Scientific Research on Innovative Areas (Research in a proposed research area) No. JP20H04631, No. JP20H05153, No. JP19H05825 and No. JP19H04683; the Grant-in-Aid for Research Activity start-up No. JP20K22476), JST CREST (No. JPMJCR20C1 and No. JPMJCR20T2), and MEXT (Innovative Area "Nano Spin Conversion Science" (No. 26103005)). Y.C., Y.T., and K.O. are supported by GP-Spin at Tohoku University. Y.C. is also supported by the Japan Society for Promotion of Science through a research fellowship for young scientists (No. JP18J21304).

## Author contributions

Y.C. designed the experiments in discussion with Y.S and E.S. Y.T., Y.N., T.M., and M.F. grew single crystals for this study. Y.C. and K.O. nano-fabricated devices. Y.C. collected and analyzed the experimental data. M.S. preformed the theoretical calculations. E.S. supervised this work. Y.C., M.S., Y.S., and E.S. wrote the manuscript. All authors discussed the results and commented on the manuscript.

## Competing interests

The authors declare no competing interests.
