## [Peer Review File · Nature Communications]

REVIEWER COMMENTS

Reviewer #1 (Remarks to the Author):

The authors show an interesting SSE signal occurring from the Spin-Peirls system CuGeO₃. The signal they observe increases linearly from 0 field, but then saturates near 5T, then shows a slight decrease. Comparing directly to Pt/YIG, the authors observe opposite sign spin current from CuGeO₃. The authors central conclusion from the results is that the signal is actually due to Triplon (dimerized Cu (spin ½)) which have a S = -1,0,+1 configuration. The authors point out that this state (in contrast to a regular ferromagnet with a S = -1 ground state at H > 0, the triplon state has a S = +1 ground state under H > 0. Therefore, they argue the sign of the spin current driven by thermal gradient will be opposite that of a typical magnon spin current. This is a nice explanation and is a reasonable explanation for the sign of the observed signal. Secondly, the authors nicely show that the signal emerges below the SP transition and that it disappears with mild Zn alloying (1-3%). I also agree with the authors rejection of the possibility of paramagnon signal because they have determined the density of free spins from Brillouin fits of the samples and found that the samples with higher density of free spins have less or no signal.

The central difficulty I have with accepting the interpretation is that the signal does not follow the magnetization. Looking at the M vs H data of the supplementary, we see that the SP phase shows a very clear non-linear signal at H_m at ~13T. This is a true and distinct signature of the SP phase and allows clear experimental correlation to the associated spin current. However, the SSE data presented in the paper is not shown above 5T and therefore we cannot see if there is any association with the SSE signal and the SP phase. The authors provided a theory that is able to explain the SSE vs H behavior as due to a magnetic field dependent scattering, but this theory is fairly ad hoc. In such a case, the data should stand on their own because the theory is built around the observable. The accepted method for determining SSE is to examine the M vs H dependence and correlate it with the spin-current because the spin-current is carried by thermal excitations of the magnetic system, whose populations are directly related to the susceptibility of the system. It is therefore quite necessary to show (as in all previous SSE papers) that the detected spin current is tracking the magnetization. It is therefore the key missing data. The authors must plot SSE vs H and M vs H over the same range in the same figure so we can see the correlation. Especially important is the H_m field and associated non-linear M vs H response.

I would be supportive of publication in Nature Comms if the authors can address the experimental points above.

Minor comments and additional technical questions:

1) Intro, authors state that SSE is observed in a range of materials, but I think they have left out a few other important ones: non-magnetic semiconductors, ferromagnetic semiconductors, ferromagnetic metallic alloys, antiferromagnets, superconductors. Along these lines, in the intro it isn't clear if the authors are trying to restrict discussion to magnetic insulators, in which case that should also be made clear.

2) P.4 line 69, "our theoretical calculation," which calculation?

3) P.5 lines 89-97 authors discuss how SSE signal appears just below 15K, but in the data presented this is not clear. The signal magnitude grows slowly below 5T for $T < 10\text{K}$, but above that there doesn't appear to be anything. If they want to show something, they would need to zoom in better. The signal could also be visible at higher fields perhaps (related to main point on Hm).

4) Related to the T dependence, when the system is above T_{SP} is the system a pure paramagnet? Is it an antiferromagnet? In either of these cases one would still expect to see some SSE signal.

5) p.6 line 112, authors refer to 1-3% Zn as "impurity doping" but this is a rather high doping level. I would consider this to be alloying.

6) P.8 line 143, "group velocity, relaxation time, and energy dispersion, respectively", the terms are not in the same order with their symbols. It looks like relaxation time needs to be swapped with dispersion to match the order.

7) Fig2. Authors compare CuGeO₃ at $T=15\text{K}$ and $T=2\text{K}$ to YIG at 300K. To make the experimental comparison more direct, authors should show data from YIG at $T=15\text{K}$ and $T=2\text{K}$. That would rule out any paramagnetic SSE related to Pt detector alone.

Reviewer #2 (Remarks to the Author):

The authors present a study of thermal spin transport in CuGeO₃, a well-known spin-Peierls material. They use an applied magnetic field as well as a thermal gradient. This leads to a transverse

voltage in a thin heavy metal layer that is in contact with CuGeO₃. This is certainly a very interesting observation. The authors go on to interpret their results in terms of spin currents in CuGeO₃. In their picture, the magnetic field leads to a splitting of triplon excitations, with the lower energy excitations carrying a magnetic moment along the applied field. An applied thermal gradient now leads to an imbalance in the excitation density. Consequently, this leads to a flow of triplons (spinful excitations) from the hot end to the cooler end. This spin current injects itself into the metal, where it is converted into a transverse voltage by the inverse spin Hall effect. The authors characterize the strength of the voltage signal for various values of the temperature and the magnetic field.

I have gone through the manuscript and the supplementary material carefully. The experimental results, as reported, are certainly very interesting. The presentation is largely clear, making for interesting reading.

However, I am not fully convinced about their interpretation in terms of spin currents. I request the authors to consider the following concerns:

1. Thermal currents are, by their nature, non-equilibrium phenomena. This is especially true in the setup used. Heat is pumped in at one end, with no heat sink at the other end. Even if a spin current is generated, it can only be a transient current. I do not see any discussion on this issue in the manuscript. How long are currents sustained? Do the results change if the measurement is carried out over a longer duration? The authors present data at various temperatures -- how can they be sure that the sample is indeed at that temperature?

2. In my view, the most important assumption here is that spin currents can tunnel into the metal. This is a big assumption that must be discussed in some detail. As CuGeO₃ is an insulator, it does not have free electronic carriers. The coupling between the metal and CuGeO₃ must arise from spin-dependent scattering at the interface. I only see a brief note at the end of the supplementary material. I find no details here. For example, what are the detailed assumptions that go into Eq. 17 (of the suppl. material)? It appears to me that the calculation does not assume a steady-state current. This is not consistent with the description of the phenomenon as given in the main text.

3. There are further assumptions in the analysis that should be explained. Changes in V_{SSE} are attributed to triplon scattering and triplon-triplon interactions. A different possibility is that these arise from the tunneling amplitude from CuGeO₃ to the metal. The resistivity of the metal may change for spin currents, as opposed to charge currents.

4. I feel the authors should carry out more measurements to rule out other interpretations. For example, the transverse voltage could simply be a consequence of the Nernst effect in the metal. One could argue that the neighbouring CuGeO₃ layer exerts an effective magnetic field on the metal layer. This effective magnetic field and the heat current within the metal lead to a transverse Nernst voltage. Note that there no spin currents involved in this interpretation.

5. Minor comment: I find one sentence in the abstract to be somewhat misleading: 'The triplon spin Seebeck effect persists even far below the spin-Peierls transition temperature...'

In my view, the word 'even' should be removed. I would think that the triplon current is expected to dominate when the temperature is much below the spin-Peierls transition temperature.

R. Ganesh

Reviewer #3 (Remarks to the Author):

This manuscript reports a spin-Seebeck (SSE) signal generated in CuGeO₃ in the presence of an external magnetic field at temperature below $T_S=14.5\text{K}$. Below T_S the Cu spins in this material dimerise, forming singlets with $S=0$ in the ground state. The spin excitations form $S=1$ triplet states with no net moment in the absence of field. In the presence of field, the dispersion relations of the three triplet excitations split, enabling net spin transport. This manuscript describes how this spin transport gives rise to the SSE signal. The authors also report a “null” experiment: when the samples are alloyed with 3% Zn substituted for the Cu, which destroys the triplons by inducing free Cu moments, the SSE signal goes away. Alloying 1% Zn attenuates the signal. The temperature-dependence of the SSE signal is consistent with the magnetization signature of the triplon state. The magnetic field dependence is consistent with a Boltzmann equation model for spin transport and scattering by unintentional Cu free spins in the samples. Overall, I am convinced that the authors have proven their point. I am also convinced of the importance of the result: this is, to my knowledge, the first evidence for transport of a triplon spin current. Therefore, I support publication in Nature Communication as-is.

Here follow a few points of detail for the authors to consider. Points 4 and 5 are trivial.

1. The dimerization is accompanied by a lattice distortion, indicating strong spin-phonon coupling. Yet the model for the field and temperature dependence rests mostly on triplon-free spin scattering. Shouldn't triplon-phonon scattering also be considered, e.g. in the temperature-dependence, which

would then have three factors, the triplon population, the Brillouin function describing the density of free Cu spins, and phonon scattering?

2. In refuting the Nernst effect explanation, the authors refer to its linearity in field. Much stronger arguments are its absence in the $\text{Zn}_{0.03}\text{Cu}_{0.97}\text{GeO}_3$ sample and in the CuGeO_3 sample at $T > T_s$.

3. Methods: all SSE data are anti-symmetrized. Is this justified by crystal symmetry? Could the authors add a non-symmetrized, raw-data curve to the supplement, for the reader to judge the effect of this data-treatment operation?

4. I find the ordinate scales used in Fig. 3 a-f less than transparent for the reader. The ordinate axes give only the zero; the scale is given by a scale bar. Would it be possible to give 0 and 1 $\mu\text{V}/\text{W}\cdot\Omega$ in frames a-d? There is no real reason to use this scheme in frames e and f.

5. Typo, p10 line 195: gapped has 2 p's.

Reply to Reviewer 1

Blue: Reviewer's questions/comments

Black: Response

First of all, we thank Reviewer 1 for the valuable comments to our manuscript. We have carefully addressed the reviewer's comments and revised the manuscript accordingly; all the changes in the manuscript are highlighted in red. Our responses and the corresponding revisions of the manuscript are as follows:

The authors show an interesting SSE signal occurring from the Spin-Peierls system CuGeO₃. The signal they observe increases linearly from 0 field, but then saturates near 5T, then shows a slight decrease. Comparing directly to Pt/YIG, the authors observe opposite sign spin current from CuGeO₃. The authors central conclusion from the results is that the signal is actually due to Triplon (dimerized Cu (spin 1/2)) which have a $S = -1, 0, +1$ configuration. The authors point out that this state (in contrast to a regular ferromagnet with a $S = -1$ ground state at $H > 0$, the triplon state has a $S = +1$ ground state under $H > 0$. Therefore, they argue the sign of the spin current driven by thermal gradient will be opposite that of a typical magnon spin current. This is a nice explanation and is a reasonable explanation for the sign of the observed signal. Secondly, the authors nicely show that the signal emerges below the SP transition and that it disappears with mild Zn alloying (1-3%). I also agree with the authors rejection of the possibility of paramagnon signal because they have determined the density of free spins from Brillouin fits of the samples and found that the samples with higher density of free spins have less or no signal.

(Authors' Response)

We greatly appreciate the reviewer's positive comment. We have carefully revised our manuscript in response to your comments, as shown below.

Prior to our reply to your comments, here we would respectfully comment on the triplon excitations and ground state of CuGeO₃ under external magnetic fields. As the reviewer pointed out, the $S=1$ triplon is the lowest excitation in the spin-Peierls (SP) phase of CuGeO₃ in a zero magnetic field. Even in a finite magnetic field $|H| > 0$, the dominant carrier (excitation) of spin current is given by the $S=+1$ triplon. This is because the ground state of the SP phase in magnetic fields is **an $S=0$ state**, while the degeneracy of the triplon states is lifted due to the Zeeman effect.

(Question/Comment 1)

The central difficulty I have with accepting the interpretation is that the signal does not follow the magnetization. Looking at the M vs H data of the supplementary, we see that the SP phase shows a very clear non-linear signal at H_m at $\sim 13T$. This is a true and distinct signature of the SP phase and allows clear experimental correlation to the associated spin current. However, the SSE data presented in the paper is not shown above 5T and therefore we cannot see if there is any association with the SSE signal and the SP phase. The authors provided a theory that is able to explain the SSE vs H behavior as due to a magnetic field dependent scattering, but this theory is fairly ad hoc. In such a case, the data should stand on their own because the theory is built around the observable. The accepted method for determining SSE is to examine the M vs H dependence and correlate it with the spin-current because the spin-current is carried by thermal excitations of the magnetic system, whose populations are directly related to the susceptibility of the system. It is therefore quite necessary to show (as in all previous SSE papers) that the detected spin current is tracking the magnetization. It is therefore the key missing data. The authors must plot SSE vs H and M vs H over the same range in the same figure so we can see the correlation. Especially important is the H_m field and associated non-linear M vs H response. I would be supportive of publication in Nature Comms if the authors can address the experimental points above.

(Authors' Response 1)

We thank the reviewer's comment. Following the comments, we have newly mentioned the comparison between $M-H$ and $V_{\text{SSE}}(H)$ in the main text (page 5-6, lines 102-109). Moreover, we have newly added Fig. S2 in the Supplementary Information pages 3-5 to compare the SSE data with magnetization in the field range of 0-14 T; please see the following Fig. R1.

Figure R1 / Figure S2: Comparison between $M(H)$ and $|V_{\text{SSE}}|$ at different temperatures.

In the case of ferro(i)magnets, the SSE indeed scales with the $M-H$ curve. The situation is, however, quite different in the case of the spin-Peierls (SP) phase. The magnetization of the CuGeO_3 at $T < T_{\text{SP}}$ is mainly due to the impurity spins and unpaired spins on thermally broken dimers. As T increases, the density of broken dimers rises and leads to a larger static magnetic susceptibility. In contrast, the scattering probability between triplons and broken dimers becomes larger with the increase of the broken dimers, and therefore the triplons' lifetime and mean free path become shorter with increasing T . As a result, the triplon SSE signal should decrease with increasing T . In other words, a larger static susceptibility roughly stands for more defects in the ground state (more broken dimers) in the SP phase; this is highly in contrast to ferromagnets, in which a larger magnetization means the increase in the spin-current polarization.

Moreover, we respectfully note that in the SP phase, as T increases, not only the $S=+1$ triplon density but also $S=-1$ triplon one increases, and, as a result, cancellation between spin currents carried by $S=+1$ and $S=-1$ triplons is more significant. This also contributes to the decrease in the SSE signal.

Because of the different properties between ferro(i)magnets and SP phases, the triplon SSE signal does not scale with magnetization, as shown in Fig. R1 (please see also Fig. S2 in SI). In other words, this unusual H dependence is one of the characteristics of the triplon spin current.

As shown in Fig. R1a (Fig. S2a), the SSE signal does not scale with $M-H$ at $T = 2$ K and 6 K. The $M-H$ in the low field range reflects the paramagnetic term from free (impurity) spins, and these free spins do not play a role in $V_{\text{SSE}}(H)$ because they are well decoupled. At $T = 6$ K, the magnetic component from broken dimers overwhelms the paramagnetic component from impurity spins. Since

the broken dimers are also little important for spin-current carriers in SSE, V_{SSE} and M should show totally different magnetic field dependence.

Also, the anomaly in the SSE is hardly observed at H_m . This is because the net spin current above H_m is still determined by low-energy $S=+1$ triplons and high-energy $S=-1$ triplons, similar to the case of $H < H_m$. Please see the following Fig. R2 which illustrates the triplon bands above H_m . As shown in Fig. R2, for $H > H_m$, the excitation energy of the $S=+1$ triplon is lower than the energy of the ground state, and the $S=+1$ triplon condenses. This is called the triplon BEC (*Nature Phys* **4**, 198–204, 2008). The condensed triplon shows static magnetization but does not carry a spin current.

The comparison with the SSE in antiferromagnetic insulators would be useful to deeply understand the feature of the triplon SSE. In the case of the antiferromagnet/Pt system, the SSE signal changes abruptly near the spin-flop transition [Seki, S. *et al. PRL*. **115**, 266601, 2015 and Wu, S. M. *et al. PRL*. **116**, 097204, 2016]. At the spin-flop transition, both the ground state and the magnon band structure drastically change and a net magnetization suddenly appears because the spin-flop transition is of a first-order type. Due to this significant change of the magnon band, the SSE signal largely changes at the spin-flop transition. On the other hand, although the triplon BEC occurs for $H > H_m$ of CuGeO_3 , the band structure of the triplon excitation is very similar to that for $H < H_m$. Therefore, there was no significant change in the triplon SSE signal around H_m .

Figure R2 / Figure S3: a schematic illustration of triplon BEC. Green, black and red curves represent three triplet states $|\downarrow\downarrow\rangle$, $(|\uparrow\downarrow\rangle + |\downarrow\uparrow\rangle)/2$, and $|\uparrow\uparrow\rangle$ respectively.

The reviewer recommended to show experimental results of a wide parameter range. Following the reviewer's recommendation, we newly added Fig. S2 in the revised Supplemental Information. Finally, we respectfully comment on our theoretical analysis. Our two approaches (Boltzmann equation and tunnel spin current) start from the sine-Gordon (SG) model (Eq. (3) of Supplementary Information). It is widely known that the SG model can well describe the low-energy physics of the SP phase in the quantitative level. Therefore, our theory is quite reliable at low temperatures and low magnetic fields. On the other hand, when T or H increases, (as we mentioned) the dimerized ground state is partially broken down and the SG model gradually becomes less reliable. This is the main reason why we focused on the low-temperature and low-field regimes in the previous figures. We also respectfully emphasize that the tunnel spin current formula of Eq. (18) (Eq. (17) in the former version of Supplemental Information) has succeeded in well explaining SSEs in several magnets such as a 1D spin liquid, a spin-nematic magnet, and a compensated ferrimagnet (in addition to usual ferro(i)magnets). Please see Supplementary Information Refs. [1,27,31].

(Question/Comment-2)

Minor comments and additional technical questions: 1) Intro, authors state that SSE is observed in a range of materials, but I think they have left out a few other important ones: non-magnetic

semiconductors, ferromagnetic semiconductors, ferromagnetic metallic alloys, antiferromagnets, superconductors. Along these lines, in the intro it isn't clear if the authors are trying to restrict discussion to magnetic insulators, in which case that should also be made clear.

(Authors' Response 2)

We appreciate this comment. In the revised manuscript, we have added new sentences on page 2, lines 32-35 to restrict discussion to magnetic insulators.

(Question/Comment 3)

2) P.4 line 69, "our theoretical calculation," which calculation?

(Authors' Response 3)

We apologize for this unclear expression. It is the calculation based on the tunnel spin current. In the revised version, we have specified it (page 4 lines 73-74).

(Question/Comment 4)

3) P.5 lines 89-97 authors discuss how SSE signal appears just below 15K, but in the data presented this is not clear. The signal magnitude grows slowly below 5T for $T < 10K$, but above that there doesn't appear to be anything. If they want to show something, they would need to zoom in better. The signal could also be visible at higher fields perhaps (related to main point on Hm).

(Authors' Response 4)

Thank you for the suggestion. In the revised manuscript, we have newly shown the rescaled SSE data at 2 K, 6 K, and 15 K in Fig. S2 in Supplementary Information (please see also Fig. R1). To make it clear that the SSE signal above T_{SP} is negligibly small compared to that at 2 K, we do not rescale Fig. 2 in the main text.

(Question/Comment 5)

4) Related to the T dependence, when the system is above T_{SP} is the system a pure paramagnet? Is it an antiferromagnet? In either of these cases one would still expect to see some SSE signal.

(Authors' Response 5)

Thank you very much for the insightful comment. Following the comment, in the revised version, we added Supplementary Information page 4, lines 36 – 44.

Above T_{SP} , the system is theoretically predicted to be a type of paramagnets: a Tomonaga-Luttinger (TL) spin liquid, where the equilibrium state is still a zero magnetized state in the absence of fields and the spin excitation is a gapless spinon instead of triplon. For higher temperatures ($k_B T > J$), the system experience a crossover from the TL spin liquid to a paramagnetic state. As the reviewer commented, we also expect a spinon SSE (Hirobe, D. *et al*, *Nature Physics* **13**, 30–34, 2017) in the intermediate temperature range ($T_{sp} < T \ll J/k_B$).

From the paper by Hirobe, *et al*, the spinon SSE signal in TL spin liquids was found to be much smaller than the ferromagnetic SSE, linear with respect to the magnetic field. The SSE signal measured at 15 K $> T_{SP}$ satisfies all these properties, and thus may be an indication of the spinon SSE. Spinon SSE in CuGeO_3 is also a very interesting future topic, but since we are focusing on the triplon SSE in this paper, we did not discuss it further.

(Question/Comment 6)

5) p.6 line 112, authors refer to 1-3% Zn as "impurity doping" but this is a rather high doping level. I would consider this to be alloying.

(Authors' Response 6)

We have deleted the word "impurity" and changed it to "Zn doping" in the main text; please see page 7 lines 131-132, and page 8 lines 147, 148, 157.

(Question/Comment 7)

6) P.8 line 143, “group velocity, relaxation time, and energy dispersion, respectively”, the terms are not in the same order with their symbols. It looks like relaxation time needs to be swapped with dispersion to match the order.

(Authors’ Response 7)

Thank you very much for pointing out the typo. We have corrected the order of them on page 8 line 161.

(Question/Comment 8)

7) Fig2. Authors compare CuGeO₃ at T=15K and T=2K to YIG at 300K. To make the experimental comparison more direct, authors should show data from YIG at T=15K and T=2K. That would rule out any paramagnetic SSE related to Pt detector alone.

(Authors’ Response 8)

Thank you very much for your suggestion. We have replaced the SSE data of YIG at 300 K by the new data measured at 2 K and 15 K in Fig. 2.

Reply to Reviewer 2

Blue: Reviewer's questions/comments

Black: Response

First of all, we thank Reviewer 2 for the valuable comments on our manuscript. We have carefully addressed the reviewer's comments and revised the manuscript accordingly; all the changes in the manuscript are highlighted in red. Our responses and the corresponding revisions of the manuscript are as follows:

The authors present a study of thermal spin transport in CuGeO₃, a well-known spin-Peierls material. They use an applied magnetic field as well as a thermal gradient. This leads to a transverse voltage in a thin heavy metal layer that is in contact with CuGeO₃. This is certainly a very interesting observation. The authors go on to interpret their results in terms of spin currents in CuGeO₃. In their picture, the magnetic field leads to a splitting of triplon excitations, with the lower energy excitations carrying a magnetic moment along the applied field. An applied thermal gradient now leads to an imbalance in the excitation density. Consequently, this leads to a flow of triplons (spinful excitations) from the hot end to the cooler end. This spin current injects itself into the metal, where it is converted into a transverse voltage by the inverse spin Hall effect. The authors characterize the strength of the voltage signal for various values of the temperature and the magnetic field.

I have gone through the manuscript and the supplementary material carefully. The experimental results, as reported, are certainly very interesting. The presentation is largely clear, making for interesting reading.

However, I am not fully convinced about their interpretation in terms of spin currents. I request the authors to consider the following concerns:

(Authors' Response)

We thank the reviewer for your positive comments on our manuscript. We carefully address your concerns as follows.

(Question/Comment 1)

Thermal currents are, by their nature, non-equilibrium phenomena. This is especially true in the setup used. Heat is pumped in at one end, with no heat sink at the other end. Even if a spin current is generated, it can only be a transient current. I do not see any discussion on this issue in the manuscript. How long are currents sustained? Do the results change if the measurement is carried out over a longer duration? The authors present data at various temperatures -- how can they be sure that the sample is indeed at that temperature?

(Authors' Response 1)

In response to this comment, we have newly discussed this issue in Supplemental Information pages 8-9, lines 87-96.

Since AC heat current is *continuously* applied in our measurement, the spin current is not a transient current. Another problem may be whether the ac spin current follows the ac heat current with 26 Hz. According to the time-resolved SSE for magnon systems (Agrawal, M. *et al. Phys. Rev. B* **89**, 224414, 2014), it was found that the SSE signal follows the heating impulse in a microsecond time scale. The frequency of heat current in our measurement is set at 26 Hz. One period of heat current (38 ms) is sufficiently long for the spin system to reach a non-equilibrium steady state, since the spin relaxation time is much shorter than this time period. That is, our experimental setup observes the thermally generated "pseudo-DC" spin current. We also confirmed that the signal does not change in the frequency range from 3Hz to 100Hz. This also indicates that our measurement is performed over a much longer time period than the spin relaxation.

We then discuss the temperature stability in our SSE measurement. In our experimental setup, the on-chip device is fabricated on a large CuGeO₃ sample, which is mounted onto a PPMS rotator chip. The chip is connected to the PPMS chamber (heat bath). The heating power of the Au-film heater

(~ 0.05 mW) is much smaller than the cooling power of PPMS at 2 K (\sim mW). Therefore, we consider the backside of the CuGeO₃ is held at the base temperature of PPMS.

As for the actual temperature at the interface, we can use the Pt detector layer as a thermometer (i.e. Pt resistance thermometer) to investigate the temperature change with and without the Au heating current (e.g. Wu, S. M. et al. *J. Appl. Phys.* **117**, 17C509, 2015). When the heater current is applied to the Au heater, the resistance of the Pt detector changes, as shown in Fig. R3a (please see Fig. S5 in SI). Using dR/dT and $\Delta R \equiv R(I_{\text{Au}} = 1.5 \text{ mA}) - R(I_{\text{Au}} = 0)$, we estimated the typical temperature difference across the device $\Delta T = \Delta R / (dR/dT)$ to be 0.2-0.5 K, as shown in Fig. R3(S5)d. For $T > 5$ K, ΔT is as small as 0.25 K and 0.5 K even at $T = 2$ K.

Figure R3 (Figure S6 in SI): a, Temperature dependence of the resistance of Pt with and without heating. b-d, Temperature dependence of dR/dT , ΔR , and ΔT . The dotted curve in d is a guide for the eyes.

(Question/Comment 2)

2. In my view, the most important assumption here is that spin currents can tunnel into the metal. This is a big assumption that must be discussed in some detail. As CuGeO₃ is an insulator, it does not have free electronic carriers. The coupling between the metal and CuGeO₃ must arise from spin-dependent scattering at the interface. I only see a brief note at the end of the supplementary material. I find no details here. For example, what are the detailed assumptions that go into Eq. 17 (of the suppl. material)? It appears to me that the calculation does not assume a steady-state current. This is not consistent with the description of the phenomenon as given in the main text.

(Authors' Response 2)

We greatly appreciate the reviewer's comment. As the reviewer pointed out, the interface interaction between CuGeO₃ (magnetic insulator) and Pt (heavy metal) is very important for the spin current to be injected from CuGeO₃ into Pt. Following the reviewer's comment, we have newly added some explanations (Supplemental Information pages 17-19, lines 226-236; lines 252-261) about the interface interaction and the derivation of Eq. (17) (the label is Eq. (18) in the revised version of Supplemental Information).

In the derivation of Eq. (18), we have assumed that the dominant interaction at the interface is given by Heisenberg-type exchange interaction, which has been used to describe interfacial spin transfer between an insulator magnet and a metal (Hirobe, D. *et al*, *Nature Physics* **13**, 30–34, 2017 & Adachi, H. *et al*, *Phys. Rev. B* **83**, 533, 2011), between the localized spin of CuGeO₃ and conducting-electron spin of Pt. We can arrive at the spin current formula of Eq. (18) by applying the perturbation theory with respect to the interface exchange interaction J_{int} . We stress that if exchange interaction exists at the interface, a spin current can be injected from CuGeO₃ to Pt even without conducting electron tunneling at the interface. In the perturbation theory based on Keldysh Green's function, we have also assumed that the nonequilibrium steady state is realized by a long-time application of thermal gradient. Thus Eq. (18) stands for the tunnel spin current in the nonequilibrium steady state driven by a thermal gradient. We respectfully emphasize that this formula can be applied to a broad class of bilayer

systems consisting of a magnet and a paramagnetic metal because we do not use any assumption about the sorts of the magnet in the derivation. In fact, the spin current formula of Eq. (18) has succeeded in well explaining the SSE in a 1D spin liquid (Sr_2CuO_3), a spin-nematic magnet (LiCuVO_4) and a compensated ferrimagnet (GdIG) on top of the usual ferro(i)magnets. Please see Supplementary Information Refs. [1,27,31]. These results strongly indicate that our assumption of an interface exchange interaction is reasonable in a wide class of SSE set up consisting of a magnetic insulator and a paramagnetic metal.

We also guess that reviewer 2 might be concerned about what is the thing that tunnels from CuGeO_3 into Pt. In the case of a bilayer junction between two metals (metals A and B), as reviewer 2 pointed out, electrons in metal A can tunnel to metal B and vice versa. On the other hand, triplons in CuGeO_3 cannot be injected into the metal Pt. However, we note that the spin of triplons can be transferred to the spin of conducting electrons in metals through the exchange interaction J_{int} at the interface. Namely, “spin angular momentum” can tunnel from the magnetic insulator CuGeO_3 to the metal Pt via the interface exchange interaction (although any quasi-particle does not tunnel into the metal). We added this meaning of “tunneling” in section G3 of the Supplemental Information.

(Question/Comment 3)

3. There are further assumptions in the analysis that should be explained. Changes in V SSE are attributed to triplon scattering and triplon-triplon interactions. A different possibility is that these arise from the tunneling amplitude from CuGeO_3 to the metal. The resistivity of the metal may change for spin currents, as opposed to charge currents.

(Authors' Response 3)

We would like to thank the reviewer's comment. We have newly discussed this issue on Supplemental Information page 19, lines 258-261. The T and H dependence of the interface spin exchange is negligible because it is governed by the crystal structure and chemical bonds at the interface, which are usually stable and almost unchanged in the present range of T and H (0-15K and 0-14T). Namely, the energy scale of the interface chemical bonds is usually expected to be much higher than those of T and H . Under the assumption of the existence of the interface exchange J_{int} and its stability, we derive the formula of Eq. (18), starting from the microscopic Hamiltonian. Therefore, we emphasize that it is NOT necessary to introduce additional phenomenological T or H dependent tunneling amplitude in our formula of Eq. (18). We would like to again stress that our microscopic formula of Eq. (18) has succeeded in explaining the SSE experiments of a ferromagnet, a compensated ferrimagnet, a 1D spin liquid, and a spin-nematic magnet.

Finally, we would respectfully note that the T and H dependence of the Pt resistivity is experimentally studied in Fig. S4 in the Supplementary Information. The resistivity is almost constant for T and H , indicating that the T and H dependence of resistivity is irrelevant to the observed signal.

(Question/Comment 4)

4. I feel the authors should carry out more measurements to rule out other interpretations. For example, the transverse voltage could simply be a consequence of the Nernst effect in the metal. One could argue that the neighbouring CuGeO_3 layer exerts an effective magnetic field on the metal layer. This effective magnetic field and the heat current within the metal lead to a transverse Nernst voltage. Note that there no spin currents involved in this interpretation.

(Authors' Response 4)

Following the reviewer's instruction, we carried out some measurements. We have newly added the experimental results on the (001) surface in the main text page 7, lines 125-131 and Figs. 3c and 3h. On page 10, lines 199-202 of the main text were revised.

As shown in the following Fig. R4c, we have newly performed reference experiments in a different crystal orientation: the Pt detector is fabricated on the (100) surface. In this case, the spin chain is parallel to the surface, and the injection of spin current is not expected. As a result, under the same experimental setup as the main text, the SSE signal is reduced to about one-twenty of that measured along the spin chain direction. This result supports that the observed signal indeed comes from the

1D triplon transportation. If the signal originated from the Nernst effect, the signal should be larger in the new setup (the (100) surface of CuGeO_3 is cleaved and has a clearer surface than the cut-and-polished (001) surface).

The effective magnetic field from the CuGeO_3 substrate plays a minor role in our experiment because the signal disappears in the high temperature spin liquid phase and Zn-doped sample, in which the magnetization is larger than the spin-Peierls phase.

Figure R4 (new Figure 3 in the main text): T dependence of V_{SSE} . **a** and **b**, H dependence of V_{SSE} at selected temperatures for CuGeO_3/Pt , with ∇T applied along the spin chain. **c**, H dependence of V_{SSE} for CuGeO_3/Pt , with ∇T applied perpendicular to the spin chain. **d** and **e**, H dependence of V_{SSE} at selected temperatures for $\text{Cu}_{0.99}\text{Ge}_{0.01}\text{O}_3/\text{Pt}$ and $\text{Cu}_{0.97}\text{Ge}_{0.03}\text{O}_3/\text{Pt}$, respectively. In **a-e**, the SSE signal is normalized by the heating power and the detector resistance. **f** and **g**, T dependence of magnetic susceptibility for $\text{Cu}_{0.99}\text{Ge}_{0.01}\text{O}_3/\text{Pt}$ and $\text{Cu}_{0.97}\text{Ge}_{0.03}\text{O}_3/\text{Pt}$, respectively. **h**, T dependence of SSE signal in different samples at $\mu_0 H = 2.4$ T.

(Question/Comment 5)

5. Minor comment: I find one sentence in the abstract to be somewhat misleading: 'The triplon spin Seebeck effect persists even far below the spin-Peierls transition temperature...' In my view, the word

'even' should be removed. I would think that the triplon current is expected to dominate when the temperature is much below the spin-Peierls transition temperature.

(Authors' Response 5)

Thank you for your suggestion. Following this comment, we have removed the word *'even'* from the abstract (page 2, line 26).

Reply to Reviewer 3

Blue: Reviewer's questions/comments

Black: Response

First of all, we thank Reviewer 3 for the valuable comments to our manuscript. We have carefully addressed the reviewer's comments and revised the manuscript accordingly; all the changes in the manuscript are highlighted in red. Our responses and the corresponding revisions of the manuscript are as follows:

This manuscript reports a spin-Seebeck (SSE) signal generated in CuGeO₃ in the presence of an external magnetic field at temperature below $T_S=14.5\text{K}$. Below T_S the Cu spins in this material dimerise, forming singlets with $S=0$ in the ground state. The spin excitations form $S=1$ triplet states with no net moment in the absence of field. In the presence of field, the dispersion relations of the three triplet excitations split, enabling net spin transport. This manuscript describes how this spin transport gives rise to the SSE signal. The authors also report a "null" experiment: when the samples are alloyed with 3% Zn substituted for the Cu, which destroys the triplons by inducing free Cu moments, the SSE signal goes away. Alloying 1% Zn attenuates the signal. The temperature-dependence of the SSE signal is consistent with the magnetization signature of the triplon state. The magnetic field dependence is consistent with a Boltzmann equation model for spin transport and scattering by unintentional Cu free spins in the samples. Overall, I am convinced that the authors have proven their point. I am also convinced of the importance of the result: this is, to my knowledge, the first evidence for transport of a triplon spin current. Therefore, I support publication in Nature Communication as-is.

(Authors' Response)

We appreciate your recommendation of acceptance and also many helpful comments. We have revised our manuscript based on your comments accordingly as follows.

(Question/Comment 1)

1. The dimerization is accompanied by a lattice distortion, indicating strong spin-phonon coupling. Yet the model for the field and temperature dependence rests mostly on triplon-free spin scattering. Shouldn't triplon-phonon scattering also be considered, e.g. in the temperature-dependence, which would then have three factors, the triplon population, the Brillouin function describing the density of free Cu spins, and phonon scattering?

(Authors' Response 1)

We thank the Reviewer for this valuable comment. We have newly discussed this issue in the main text page 6 lines 117-119, and in Supplemental Information pages 21-22 and Fig. S8.

As the Reviewer pointed out, we need to incorporate an additional relaxation term due to the phonon-triplon scatterings to evaluate precisely the temperature dependence of SSE. To check the contribution of phonon scattering, the temperature dependence of two parameters is required: relaxation time of phonon scattering and excitation gap of triplons. First, the relaxation time of phonon scattering can be related to the T -dependence of the lifetime of triplons $\tau(T)$ estimated from the T -dependence of the linewidth of the neutron spectrum $\Gamma(T) \propto \tau(T)^{-1}$. Experimental results of $\Gamma(T)$ [Regnault, L. P. *et al*, *Phys. Rev. B* **53**, 5579–5597, 1996] are well fitted by $\Gamma(T) \propto T^4$ in the moderate temperature range of 5 K to 10 K and thus we deduce that the T -dependent relaxation also has this form. Second, the T -dependence of the excitation gap of triplons can also be estimated from another neutron scattering result [Lussier, J. G. *et al*, *Journal of Physics: Condensed Matter* **8**, L59–L64, 1996]. The gap is well fitted as $\Delta(T) = \Delta(0)(1 - T/T_{SP})^{0.12}$.

Figure R5 / Figure S8: **a-c** Comparison between calculation results of the T dependence of the triplon spin current and the observed SSE signal at $\mu_0 H = 1, 3, 5$ T, respectively.

The calculated result of the T -dependent SSE is shown in Fig. R5 (Fig. S7 in Supplementary Information). We found a qualitative agreement between experiment and calculation in the moderate temperature range (approximately from 5 K to 10 K). As the reviewer commented, phonon-triplon scattering is important in our results.

Finally, we respectfully emphasize that the result of the temperature dependence is justified only in the moderate temperature range. The reason why our result is less reliable in the high- T range ($T \rightarrow T_{SP}$) is as follows: our theory is based on the Sine-Gordon model, which can well describe the low-energy physics of the SP phase. When T or H increases, the dimerized ground state is partially broken down and the SG model gradually becomes less reliable. This is also why theoretical calculation in Supplementary Information G1-G3 focused only on the low temperatures and low magnetic fields, in which the singlet ground state and triplon excitation are well established. On the other hand, the relation of $\Gamma(T) \propto T^4$ is not experimentally supported in the low- T range. Thus, Fig. R5 shows the temperature range from 5K to 11K.

(Question/Comment 2)

2. In refuting the Nernst effect explanation, the authors refer to its linearity in field. Much stronger arguments are its absence in the $Zn_{0.03}Cu_{0.97}GeO_3$ sample and in the $CuGeO_3$ sample at $T > T_s$.

(Authors' Response 2)

We thank you for the insightful comment. Following the comment, we have added some sentences to discuss the Nernst effect on page 10, lines 199-202 in the main text.

(Question/Comment 3)

3. Methods: all SSE data are anti-symmetrized. Is this justified by crystal symmetry? Could the authors add a non-symmetrized, raw-data curve to the supplement, for the reader to judge the effect of this data-treatment operation?

(Authors' Response 3)

In response to the comment, we have replaced all SSE data in the main text by the raw data (Figs. 2c-f, Figs. 3a-e and Figs. 4a).

In Fig. R6, we show an example of the raw voltage data. It is clear that H -even components are almost zero in the entire-field range. Conventionally, the SSE signal is anti-symmetrized to eliminate constant offsets in the measurement. In the present study, the symmetry (H -even) component is so small that the anti-symmetrized data is almost identical with the raw data, and thus the anti-symmetrization is not necessary.

(Question/Comment 4)

4. I find the ordinate scales used in Fig. 3 a-f less than transparent for the reader. The ordinate axes give only the zero; the scale is given by a scale bar. Would it be possible to give 0 and 1 $\mu\text{V}/\text{W}\cdot\text{Ohm}$ in frames a-d? There is no real reason to use this scheme in frames e and f.

(Authors' Response 4)

Thank you for your suggestion. We have newly modified the scales in Figs. 2c-f and Figs. 3a-g to make them easier to read.

Figure R6: **a**, A typical raw voltage signal in the SSE measurements for CuGeO_3/Pt at 2K. **b**, H -odd component of the signal. **c**, H -even component of the signal.

(Question/Comment 5)

5. Typo, p10 line 195: gapped has 2 p's.

(Authors' Response 5)

Thank you for pointing out the typo. We have corrected the word "gapped" on line 217, page 11.

REVIEWERS' COMMENTS

Reviewer #1 (Remarks to the Author):

The authors have carefully addressed each of my previous concerns. I support publication of the article as is.

Reviewer #2 (Remarks to the Author):

I appreciate the authors' detailed response to comments from the three referees. As far as I can see, the authors have addressed all comments satisfactorily. I am happy to recommend publication.

R. Ganesh

Reviewer #3 (Remarks to the Author):

I have read the authors' rebuttal and the revised supplement. The new discussion of the absence of relation between magnetization and SSE signal is a necessary addition. Figs S2 and S3 make the point clear. Also useful is the discussion of the state of the system at $T > T_{SP}$. This concludes my review: as far as I am concerned, the paper can be published in its present form.

I also took a look at the other reviewers' comments.

The authors are correct to point out, in reply to reviewer-2's comments, that the thermal currents and thus the thermally-driven triplon current are out of equilibrium but steady state currents. Another question from reviewer-2 is, I think, worth answering in detail: how does the spin-polarization of the electrons in the Pt come about? This is the equivalent of the concepts of spin-transfer-torque and spin-pumping at the insulating FM/NM interface. The reader would be well served if the paper contained a discussion of the analogies/differences between that system and the present one.

Reply to Reviewer 1

Blue: Reviewer's questions/comments

Black: Response

The authors have carefully addressed each of my previous concerns. I support publication of the article as is.

(Authors' Response)

We thank the reviewer for the very positive conclusion on our results and the recommendation for publication.

Reply to Reviewer 2

Blue: Reviewer's questions/comments

Black: Response

I appreciate the authors' detailed response to comments from the three referees. As far as I can see, the authors have addressed all comments satisfactorily. I am happy to recommend publication.

R. Ganesh

(Authors' Response)

Thank you once again for your valuable comments and suggestions, as well as the recommendation for publication.

Reply to Reviewer 3

Blue: Reviewer's questions/comments

Black: Response

I have read the authors' rebuttal and the revised supplement. The new discussion of the absence of relation between magnetization and SSE signal is a necessary addition. Figs S2 and S3 make the point clear. Also useful is the discussion of the state of the system at $T > T_{SP}$. This concludes my review: as far as I am concerned, the paper can be published in its present form.

(Authors' Response)

Once again, thank you for your insightful comments and for recommending our paper.

(Reviewer's comment)

I also took a look at the other reviewers' comments. The authors are correct to point out, in reply to reviewer-2's comments, that the thermal currents and thus the thermally-driven triplon current are out of equilibrium but steady state currents. Another question from reviewer-2 is, I think, worth answering in detail: how does the spin-polarization of the electrons in the Pt come about? This is the equivalent of the concepts of spin-transfer-torque and spin-pumping at the insulating FM/NM interface. The reader would be well served if the paper contained a discussion of the analogies/differences between that system and the present one.

Thank you for the comment. Reviewer 3 is concerned about the polarization direction of the spin current in metal (Pt). To clearly explain this feature, we added several sentences in Lines 77-81 in the main text.

At the interface between magnets and metal, an exchange interaction is believed to be most dominant. This is well supported by various experimental and theoretical studies of SSEs such as SSEs of ferromagnets, ferrimagnets, 1D spin liquids, antiferromagnets, spin-nematic magnets, etc. The exchange interaction conserves spin angular momentum, and therefore the polarization direction of the spin current in metal (Pt) is the same as that in magnets. Thus, the spin current in magnets determines the sign of ISHE.

As Reviewer 3 pointed out, the microscopic origins of angular momentum transfer in spin transfer-torque and spin-pumping phenomena are interesting and important. However, our present study has focused on the "thermally generated" spin current. If we add a discussion about spin-transfer torque and spin-pumping in the main text, the main purpose of our study would become dim. Therefore, we added only the explanation about the polarization direction of the spin current in Pt, while we did not touch spin-transfer torque and spin-pumping in Lines 77-81.